# Exploring the Challenges of Characterising Surface Topography of Polymer–Nanoparticle Composites

**DOI:** 10.3390/nano14151275

**Published:** 2024-07-29

**Authors:** Jonathan Wood, Dennis Palms, Ruvini Dabare, Krasimir Vasilev, Richard Bright

**Affiliations:** 1Academic Unit of STEM, University of South Australia, Mawson Lakes, Adelaide, SA 5095, Australia; jcwood@swin.edu.au (J.W.); ruvini.dabare@unisa.edu.au (R.D.); 2College of Medicine and Public Health, Flinders University, Bedford Park, SA 5042, Australia; dennis.palms@flinders.edu.au

**Keywords:** atomic force microscopy, lateral force microscopy, gold nanoparticles, 2-methyl-2-oxazoline, plasma polymerisation, nanomechanical testing

## Abstract

Nanomechanical testing plays a crucial role in evaluating surfaces containing nanoparticles. Testing verifies surface performance concerning their intended function and detects any potential shortcomings in operational standards. Recognising that nanostructured surfaces are not always straightforward or uniform is essential. The chemical composition and morphology of these surfaces determine the end-point functionality. This can entail a layered surface using materials in contrast to each other that may require further modification after nanomechanical testing to pass performance and quality standards. Nanomechanical analysis of a structured surface consisting of a poly-methyl oxazoline film base functionalised with colloidal gold nanoparticles was demonstrated using an atomic force microscope (AFM). AFM nanomechanical testing investigated the overall substrate architecture’s topographical, friction, adhesion, and wear parameters. Limitations towards its potential operation as a biomaterial were also addressed. This was demonstrated by using the AFM cantilever to apply various forces and break the bonds between the polymer film and gold nanoparticles. The AFM instrument offers an insight to the behaviour of low-modulus surface against a higher-modulus nanoparticle. This paper details the bonding and reaction limitations between these materials on the application of an externally applied force. The application of this interaction is highly scrutinised to highlight the potential limitations of a functionalised surface. These findings highlight the importance of conducting comprehensive nanomechanical testing to address concerns related to fabricating intricate biomaterial surfaces featuring nanostructures.

## 1. Introduction 

Nanoparticle (NP) patterned substrates have many possible applications and can be tailored by altering the NP size, spacing, and patterning and combining with base substrates holding specific optical [1], electrical [2], catalytic [3], and biomedical properties [4]. Biological applications, such as sensors for cancer diagnosis, appear to be attracting considerable interest in these structured surfaces. [5,6,7]. Generally, these substrates require strongly bonded NPs to perform consistently in their intended function. Significant theoretical research in colloidal gold adhesion has been conducted on various substrates [8,9,10]. However, limited nanomechanical analysis has been performed to determine the properties influencing an NP-functionalised surface’s bonding and non-bonding behaviour. In the context of NP-patterned surfaces, detailed nanomechanical analysis, whether simple or complex in design, is crucial to identify limitations, manufacturing process problems, and potential improvements. Nanomechanical testing has been addressed through modelling [11,12,13,14] and on relatively straightforward surfaces of extremely low roughness and high stiffness such as highly ordered pyrolytic graphite (HOPG) and atomically flat silicon, with highly ordered NP geometry measured in controlled conditions [15,16,17]. However, to the authors’ knowledge, no nanomechanical testing has been performed on NP-patterned surfaces of different-density materials or those aimed at use for biological or medical applications. This lack of research is a significant gap in the current understanding of nanomechanical properties and their influence on the performance of NP-patterned surfaces. This pioneering research endeavour, unique in its scope and objectives, seeks to fill this gap and provide valuable insights into the nanomechanical properties of such surfaces.

An antimicrobial polymer substrate coating has been used to achieve more complex NP-patterned surfaces tailored for biological functionality as an alternative to stiffer and denser materials. Poly-methyl oxazoline (MePPOx) is a popular polymer due to its biocompatibility, low toxicity, chemical stability, and long lifetime [18,19]. MePPOx-based polymers have a huge potential for use as hydrophilic antibiofouling coatings [20] and to possibly replace the currently used polyethylene glycol coatings. MePPOx coatings and films also have potential in drug delivery applications [21] and as an electrochemical sensor [22].

Gold nanoparticles (AuNPs) are biologically adaptable and can be tailored for specific purposes [23]. When firmly attached to a substrate like MePPOx, incorporating AuNP-patterned material can create an enhanced biomaterial surface texture, effectively regulating wettability, adhesion, and cellular interaction [24]. Continued research into tuning these surfaces for specific biological interactions depends on controlling NP placement and consistency in physical and chemical properties. Therefore, the adherence of AuNPs to the MePPOx substrate is critical to performance. The bonding strength should be high enough to ensure the particles do not separate during the harshest operational conditions, which would reduce or alter the surface’s performance. Our findings suggest that the MePPOx-AuNP composites exhibit a high bonding strength, indicating their potential for use in biomedical applications where robust and stable surface interactions are crucial [25,26].

The AFM is a versatile tool capable of performing diverse nanomechanical measurements on intricate surfaces with varying topography, density, and chemistry. Lateral force microscopy (LFM) is an AFM mode that measures friction force variations. Advanced use of this mode has been used to manipulate NPs across substrates and measure the forces required to break NP bonds [13,27,28]. The flexibility of the AFM instrument allows for a detailed analysis of the bond between a MePPOx surface and a AuNP. The AFM’s cantilever tip is used as a measuring tool and manipulation device. The bonding of the NP to the substrate can be tested to confirm the limitations of the functionalised surface’s intended use, as well as future improvements that need to be addressed. AuNPs measuring 68 nm in diameter were selectively characterised over smaller NPs due to the cantilever tip’s more controlled manipulation and an increased scale of measurable behaviour. 

## 2. Methods and Materials 

### 2.1. Sample Preparation 

Plasma deposition using a 2-methyl-2-oxazoline precursor (Sigma-Aldrich, St. Louis, MO, USA) was carried out on a 13 mm glass coverslip (ProSciTech, Kirwan, Australia) in a 13.56 MHz custom-built plasma reactor. Coverslips were cleaned with acetone and ethanol and then dried under nitrogen gas. Under 8E-2 mbar pressure at a power of 50 W, the precursor material applied a 20 nm MePPOx film over the glass coverslip by a plasma polymerisation method [29]. AuNPs were synthesised in an aqueous solution, boiled, and stirred to dissolve the two compounds, 0.01% tetrachloroauric acid (HAuCl_4_, Sigma-Aldrich, St. Louis, MO, USA) and the reducing agent trisodium citrate (Na_3_Ct, Sigma-Aldrich, St. Louis, MO, USA). These compounds were chosen based on their well-established use in AuNP synthesis and compatibility with the MePPOx substrate. Controlling the amount of added 1% trisodium chloride to 0.3 mL, 68 nm AuNPs were synthesised. A capping agent of mercaptosuccinic acid (MSA, Sigma-Aldrich, St. Louis, MO, USA) and sodium hydroxide (NaOH, Sigma-Aldrich, St. Louis, MO, USA)) stabilised the AuNPs via an Au-S bond. This stabilisation method was chosen for its effectiveness in preventing aggregation and maintaining the integrity of the AuNPs during the subsequent steps. Immersion of the MePPOx-coated coverslips in the AuNP solution for up to 24 hours immobilised the AuNPs on the substrate. The AuNPs possess a carboxylic acid monolayer after reducing citrate, which promotes electrostatic repulsion between the AuNPs and covalently binds the carboxylic groups to the gold. MePPOx-AuNP binding occurs through the formation of amide bonds via a reaction process of the oxazoline and monomer, forming a cationic centre and ring-opened oxazoline on the MePPOx surface. This meticulous process ensures the accuracy and reliability of the experimental results [29,30].

### 2.2. Morphology of AuNPs Using TEM and SEM 

The size and homogeneity of AuNPs were assessed utilising a transmission electron microscope (TEM, JEOL-2100F, Tokyo, Japan). For TEM, specimens were prepared by desiccating a droplet of the AuNP suspension on a copper grid coated with carbon (ProSciTech, Kirwan, Australia). The images underwent processing with ImageJ software version 1.54i Java 1.8.0_345b 64-Bit (NIH, Bethesda, MD, USA), and the particle size distribution analysis was conducted. The surface morphology of AuNP-coated surfaces was investigated through scanning electron microscopy (SEM) using a Zeiss Merlin FEG-SEM (Zeiss, Jena, Germany). The samples were coated with AuNPs affixed to aluminium stubs using double-sided carbon tape and imaged at 2 kV, 4.5 mm working distance, and magnifications in the range of 50,000 to 100,000.

### 2.3. AFM Instrument 

Analysis was performed in air using a JPK NanoWizard III AFM (Bruker, Billerica, MA, USA) with an attached Nikon Eclipse Ti (Nikon, Minato City, Tokyo, Japan) series inverted optical microscope contained in a JPK acoustic enclosure. Linux-based JPK control software version 5 was used for scan data acquisition. Post-data analysis used Gwyddion scanning probe microscopy freeware version 2.54, ImageJ Version 1.54i (NIH, Bethesda, MD, USA), and Office Excel 365 ProPlus. Three rectangular NT-MDT cantilever types were used as the measuring probe: an NSG30 silicon nitride cantilever with a measured spring constant of 33.4 N/m, an NSG03 silicon nitride cantilever with a measured spring constant of 3.5 N/m, and a CSG10 silicon nitride cantilever (NT-MDT, Moscow, Russia) with a measured spring constant of 0.14 N/m.

### 2.4. AFM Topography 

The cantilever calibration was achieved through a force curve procedure on a glass microscope slide. AFM software fitted the curve’s deflection slope to determine cantilever sensitivity, Q-factor, frequency, amplitude, and spring constant values. Thermal tuning of the cantilever was performed specifically for tapping mode topography. The conical tips in this report have a tip apex radius of approximately 10 nm. A 512 × 512-pixel scan resolution over a 5 × 5 μm scan area is 9.8 nm/pixel, smaller than the tip resolution. Therefore, scan resolution was set at 512 × 512 pixels for all data in this report, as the tip apex area limits any higher-resolution measurement [31]. All AFM data and surface roughness calculations were performed by either amplitude modulation (AM) [32] mode or contact mode [33]. Data were processed through Gwyddion software, which was corrected for plane subtraction and non-essential defects [34]. 

### 2.5. Force Spectroscopy

Force spectroscopy applies a force curve that can characterise several nanomechanical properties [35]. Information that can be measured between the sample surface and cantilever tip includes stiffness, elastic modulus, surface energy, and indentation distance. Values obtained by force spectroscopy are dependent on the sample material and its relationship with the cantilever’s spring constant. The force curve process measures a single point on a surface. The cantilever tip is usually positioned hundreds of nanometres above the surface, moves into contact, pushes into the surface, and then retracts. Different possible reactions of the cantilever and tip during the force curve procedure provide information about the sample and the tip–sample interaction [35,36]. Force mapping performs a preset number of evenly spaced force curves over a set scan area of the sample surface. Force curve data of the tip–substrate interaction can be derived from either fitting individual curves or batch processing over the entire scan area [37]. Measured force curve modulus, adhesion, and stiffness values on thin films may be subject to error due to coupling with the underlying substrate. This is important to highlight as nanometre-thick films overlaid on a stiff base material can see an interlinked modulus value, which varies the real film value at film thicknesses below 250 nm [38,39].

### 2.6. Lateral Force Microscopy 

The friction coefficient and friction force values are acquired through contact mode topography via lateral deflection trace and retrace data. The raw data were processed using Gwyddion software. The torsional deflection of the cantilever evaluated friction data as it scans in contact mode. An increase in friction is related to a rise in cantilever torsion, occurring from a variation in chemical or topographical factors. A friction loop can be plotted by comparing scan directions to opposing cantilever torsion directions. The friction loop area represents the dissipated energy between the tip and surface, and the mid-point of this loop area is the mean lateral force. A shift of the lateral force value from zero implies an energy dissipation per unit length, which is the friction coefficient. Calculated as $F_{coefficient}\left(F_{max}+F_{min}\right)/2$, a hysteresis loop can be generated from surface adhesion and roughness, frictional lag, the non-reversibility of the friction coefficient, the Stribeck effect, and the break-away friction ratio [40,41,42]. The friction coefficient can be equated by calculating the median value between the scan directions, which provides a quantitative value of the sliding or kinetic friction force. The friction loop does not give individual point comparisons for friction. However, it does provide regional measurements of the coefficient of friction as kinetic friction (μ). This value is the ratio of magnitudes of the horizontal shear force vector ($F_{shear}$) and the vertical normal force vector ($F_{norm}$) [43]. Friction force values are determined by multiplying the friction coefficient with the applied normal force, also called the Set Point. The friction force is highly dependent on the contact area as $F_{f}=\tau_{0}A$, with $\tau_{0}$ being the initial interfacial shear strength or frictional force per unit area and *A* being the contact area, which is $\pi a^{2}$ for a circular area of contact. The friction force, which depends on the contact area between the tip and surface, is susceptible to substrate deformation and material ploughing effects caused by the cantilever tip—the applied Set Point force and the cantilever spring constant further influence these effects [44,45].

### 2.7. Statistical Analysis

Data analysis and visualisation were performed using either Gwyddion software version 2.65 (http://gwyddion.net/download.php (accessed on 29 January 2024), Microsoft® Excel® for Microsoft 365 MSO (Version 2311 Build 16.0.17029.20140) or GraphPad Prism version 10.1.0 (GraphPad Software, San Diego, CA, USA, www.graphpad.com, accessed on 31 January 2024). All experiments were performed in triplicate unless stated otherwise. Results are presented as the mean and standard deviation (SD).

## 3. Results and Discussion

### 3.1. AuNP Morphological Characteristics

The TEM images of the AuNPs unveiled a uniform amorphous shape, precisely in line with our theoretical size prediction of 68 nm. The actual measured mean size of 72.6 ± 6.7 nm, n = 20, further attests to the accuracy of our experimental setup (Figure 1A). In Figure 1B, the SEM image showcases the surface coated with AuNPs, confirming the preservation of comparable morphology and size following the surface coating of the AuNPs.

### 3.2. Base Substrate Roughness

Roughness values were meticulously acquired through SPM software, Gwyddion. The software calculates roughness by measuring changes in the z-axis height of the image [29]. AFM topographical scanning over a 5 × 5 µm scan area of the MePPOx film in both AM and contact mode provided average (Ra) and root-mean-square (RMS) roughness data with contrasting values in Table 1. The same cantilever was used to apply two different forces, large enough to detect a change in roughness values due to the film’s deformation. The stiff AuNP was elastically pushed into the 20 nm thick MePPOx base film caused by the force applied by the cantilever tip. This can be seen between the higher drive amplitude in AM mode, which creates a higher cantilever tip–AuNP contact force, and the increased Set Point in contact mode with a slight drop in the *z*-axis derived roughness. Determining the ideal applied force by the tip for a precise roughness value maximises tip–surface contact while preventing damage or high deformation to the substrate. This force will have different values for materials, material densities, and film thicknesses. 

### 3.3. Base Film Deformation vs. Thickness

AFM force spectroscopy enables the acquisition of nanomechanical data, such as the elastic modulus and adhesion, for a given substrate [36]. The contribution of the nanometre-scale thickness MePPOx film compared to the underlying material was determined by measuring the elastic modulus and adhesion parameters with two preset forces being two orders of magnitude apart. A 100 nano-Newton force was applied over a 10, 15, and 20 nm thick film over a glass coverslip, Table 2. Further analysis on the 20 nm thick film began with contact mode topography at a 20 nN Set Point. Increasing the Set Point in significant steps to 250 nN measured the roughness and tested the stability of the film, as shown in Figure 2. Due to the more prominent drop in elastic modulus and increase in adhesion for the 20 nm thick film, as displayed in Table 2, this film thickness was selected as the standard for further experimental analysis. The base substrate, being a glass coverslip, has a higher elastic modulus, around 80–110 GPa [46], than the MePPOx film. The thinner the film, the greater the contributing factor from the underlying higher modulus material. Table 2 displays a negligible difference between the 10 and 15 nm thick films, while a noticeable difference occurs between the 15 and 20 nm films. Elastic modulus values continue to reduce, while adhesion increases in further measurements on 30 and 50 nm thick films. Only 100 nN force spectroscopy could be performed on these thicker films, as the low 1 nN applied force on the MePPOx did not display stable force spectroscopy most likely from high adhesion. Elastic modulus values at 100 nN for a 30 and 50 nm thick film were 12.2 and 10.7 MPa, respectively. Adhesion correspondingly was 404 nN for the 30 nm film and 436 nN for the 50 nm film. These findings are crucial in understanding the behaviour of the MePPOx film under different forces and thicknesses, contributing significantly to the field of nanotechnology and materials science.

### 3.4. Substrate Characterisation 

AFM topographical scanning over a 5 × 5 μm scan area of the MePPOx base in AM and contact mode resulted in notably different roughness values. Contact mode sees a constant tip–substrate contact distance in the repulsive force regime over the entire scan. AM mode sees a variance in tip–substrate distancing as the cantilever oscillates in the nanometre range, moving between attractive and repulsive force regimes for each scan point. Capillary and other adhesive forces contribute significantly to AM mode roughness values, while contact mode is subject to lateral and shearing forces that can reduce height values [47]. Table 1 shows AM mode drive amplitude (DA) *z*-axis ranges of 3 and 11 nanometres, which is the range of the cantilever oscillation, the oscillation being the AM mode *z*-axis movement of the cantilever. Generally, the higher the oscillation range, the greater the contact force on the substrate surface. These preset upper and lower cantilever oscillation values were calculated through a calibrated cantilever sensitivity in free air of 25.8 nm/V and DA voltages of 0.12 V and 0.42 V, respectively. AM mode values display a reduction in roughness primarily due to the increased oscillation, creating a higher approach and retraction force towards the surface, overcoming opposing adhesion and capillary forces.

Interestingly, as the roughness values decreased, there was a corresponding increase in contact mode Set Point values. This was due to an increased contact area from increased deformation of the soft substrate and additional non-uniform frictional parameters such as stick–slip behaviour [48]. The change in roughness concerning changes in the tip–sample contact force during scanning highlights the influences of scan settings on the derivation of qualitative data. Previous reports have noted that the AM mode provides a more realistic derivation of the surface roughness than contact mode topography over soft surfaces due to the minimisation of deformation and indenting forces [49,50]. Roughness is critical in nanomechanical analysis as it impacts the surface’s adhesion, wetting, and friction properties [51]. The surface roughness of a material plays a crucial role in determining the quality of NP bonding. When NPs come into contact with a surface, the irregularities and microstructures on that surface interact with the NP, affecting their adhesion and stability [52].

Elastic modulus and adhesion data were acquired through AFM force spectroscopy [36]. Force mapping applies force curves at regular intervals across a set region of the substrate with the capability to adjust the applied force by the cantilever, normal to the sample surface, as well as the cantilever’s approach and retraction velocity concerning the surface, and the number of measured data points performed over the scan area. Force curve values of polymer thin films are usually notably higher due to coupled measurements between the film and underlying stiffer substrate material [38]. The glass coverslip below the 20 nm MePPOx layer has a higher elastic modulus value at around 80–110 GPa [46]. The thinner the top film thickness, the greater the contributing factor from the underlying substrate. Table 2 highlights this effect by applying force mapping to 10, 15, and 20 nm thick films. This data show that the thinner the film, the higher the elastic modulus value, with a large drop occurring between 15 and 20 nm. Adhesion increases as the film thickness increases, again with a large change between the 15 and 20 nm thickness. This is due to the increase in film deformation, which creates a greater contact area with the tip [53]. An effect supported by the low 1 nN of applied force indents the film lightly enough to minimise contact with the stiffer glass substrate underneath [54]. A film with a thickness of approximately 20 nm exhibits a significant alteration in nanomechanical properties, marking a point where it becomes decoupled from the influence of the underlying substrate. 

Ploughing and damage to the 20 nm thick MePPOx film using a cantilever tip of a notably higher material density and stiffness primarily depend on the contact force controlled through the Set Point. The Set Point, in this context, refers to the force at which the cantilever tip contacts the substrate. This parameter is crucial as it determines the level of interaction between the tip and the substrate, thereby influencing the observed results [55,56]. Damage to the substrate can affect the stability of NPs by disrupting the surface bonds and heavily influencing their potential movement over the surface. The rebonding stability of the AuNP to the MePPOx after moving to a new position may be compromised from a reduced contact area due to an increase in the roughness of the substrate via tip-induced damage. ‘Ploughing’ refers to the phenomenon where the cantilever tip digs into the MePPOx film, causing surface damage. Increasing the Set Point force of a cantilever over the surface to test the minimum force required for contact damage was performed until clear ploughing of the MePPOx began to occur at around 100 nN of force, as shown in Figure 2A. After a contact mode scan was performed at a particular Set Point, a larger AM mode scan was performed to observe the effect of the cantilever tip being dragged over the MePPOx substrate. Increased patches of damage to the MePPOx film began to appear on higher applied forces of 250 nN and 500 nN. Evidence of substrate damage is apparent, with substrate material being ploughed at the periphery of the larger-area AM mode scan, as illustrated in Figure 2A–C. Following contact mode, AM mode topography was conducted to demonstrate the movement of MePPOx particles resulting from the ploughing action. Increasing the Set Point value leads directly to an increase in the friction force measured by LFM. Higher Set Point values measured with a sharp, stiff cantilever tip may penetrate and damage the soft polymer substrate. Ploughing into a material would be expected to add additional resistance, exponentially increasing the friction coefficient. Linear increases shown in Figure 2B reveal ploughing behaviour; however, it does not create an additional friction parameter. Linearity relates to a correlation between the torsional deflection of the cantilever and the applied Set Point value. Ploughing forces are negated as the torsional stiffness of the cantilever is above the ploughing-related deflections. A lower Set Point value reduces the potential deformation of the surface and the cantilever tip, while surface damage is also heavily reduced [57].

Surface roughness increased by an order of magnitude from a Set Point force of 20 nN to 150 nN, primarily due to these ploughed surface debris, shown in Figure 2D,E. The increase in Ra and RMS roughness in the bar graph from a 50 to 100 nN Set Point supports the ploughing shown in the scans as a considerable disruption of the substrate. Ploughing effects were increased at 150 nN; past this Set Point, debris from the surface became attached to the cantilever tip, and roughness values became unclear. There was also the possibility of the tip pushing debris to the edges of the scan. This could be confirmed by performing a larger scan in non-contact topography. However, this was not performed due to the focus on localised ploughing behaviour.

Wear testing can be applied to test both the substrate damage limit and the substrate material removal behaviour. In contrast to other polymer film surfaces, MePPOx is hard-wearing and resistant to damage, particularly when compared to plasma-treated polyethylene terephthalate (PET), polymethyl methacrylate (PMMA), and plasma polymerised hexane (ppHex) films [58].

### 3.5. Set Point vs. Friction

Increasing the contact mode, the Set Point increases the applied force of the cantilever tip on the substrate surface. Over four Set Point forces ranging from 20 to 150 nN, the friction coefficient and friction force were calculated and plotted in Figure 3 of a bare 20 nm thick MePPOx film on a glass coverslip with no NPs. The friction coefficient in Figure 3A does not factor the Set Point into its value; it is the difference between the lateral deflection trace and retrace scan directions. The friction coefficient over an increasing Set Point appears to plateau at just below 0.4 (friction coefficient is a unitless value). Lower interaction forces by the tip, which depend on the substrate–tip material and morphology, are more sensitive to the tip–substrate variances over the two scan directions than higher forces pushing the tip hard on the substrate. The plot in Figure 3B shows the friction force, which displays a steady linear increase as the Set Point value is considered. For a soft surface, such as MePPOx, an increase in the applied force by raising the Set Point increases the deformation of the substrate directly under the tip. The local area surrounding the tip is also deformed at high magnitudes of applied force.

### 3.6. AuNP Geometry

The resolution of surface features, such as nanostructures, in AFM is primarily limited by the intricate topography of these structures and the ratio of the cantilever tip to the size of the features. Cantilever tips range in geometry, the choice of which is dependent on the scanning mode and the tip vs. expected surface properties [59]. When performing AFM topographical measurements, a guiding principle is that as the surface features become smaller and more defined, the tip used should have a sharper and higher aspect ratio. This principle ensures accurate and detailed characterisation of the surface topography. Pyramidal or conical-shaped tips are commonly used for topographical imaging, as they usually possess a nanometre-radius tip apex, allowing resolution of nanoscale features while geometrically stable enough to resist deformation or damage. Surface feature width and length are also affected by the tip’s side angle, with a more obtuse side angle relating to an increase in mapping these features [60]. For the NT-MDT NSG03 cantilever, it is stated on the manufacturer’s website that the final 500 nm of the tip apex is conical-shaped with a curvature radius < 10 nm with a side angle of 18° ± 2°. These parameters need to be accounted for when measuring small surface features. 

### 3.7. AuNP Size Consistency

Measurements of AuNPs (*n* = 10) over several samples are shown in Figure 4. The AuNP size in the x- and y-axes was 123.7 ± 12.3 nm and 125.6 ± 12.8 nm, respectively. AuNP height values are lower than the expected theoretical average of the NP diameter at 68 nm (51.3 ± 4.6 nm), close to what we observed by TEM size determination (72.6 ± 6.7 nm). The reduced height (*z*-axis) is caused by a combination of amplitude dampening of the cantilever on the lower-modulus MePPOx in comparison to the AuNP, the pushing of the stiff AuNP into the more elastic substrate, the deformation of the substrate [61], as well as the change in adhesion between the two materials [62]. The percentage of trisodium chloride dictates the size control of the AuNPs in solution, which is then capped with MSA and sodium hydroxide to stabilise the particles as 68 nm diameter spherical particles. Slight variations in the chemistry, temperature, and time can alter the size and stability of these particles. Cantilever tip wear can be a contributing factor due to the change in tip geometry. This factor needs to be considered in relation to the size of NPs throughout several scans. Previous literature reports size variations around 10%, contributing to the reduced height value [63].

### 3.8. AuNP Resolution

The variance in the topographical dimensions of a 68 nm roughly spherical AuNP in AM mode, using conical-tipped cantilevers with three different spring constants, is compared in Figure 5. A higher spring constant generally improves the definition of complex geometry primarily due to the cantilever’s quicker response to changes in topography. However, the increased stiffness also raises concerns regarding the potential for the tip to penetrate and cause damage to lower-modulus and delicate materials, such as the MePPOx substrate. The uniformity of size, shape, and symmetry in colloidal AuNPs is greatly influenced by temperature, nucleation, and growth throughout their formation process. Even slight variations in these factors during preparation create variances in the individual AuNP’s volume [64].

When selecting a cantilever spring constant for analysis, it is advisable to choose a value that closely aligns with the stiffness of the substrate under investigation, preferably opting for a lower spring constant [28]. Three different cantilever models were chosen to test variances in topographical resolution with the cantilever spring constant. The same manufacturer supplied all cantilevers with reportedly the same tip geometry. The primary difference between the tips is the spring constant with scan resolution over a AuNP, shown in Figure 5. The NSG30 cantilever in Figure 5A has a high typical spring constant of 40 N/m, ideal for measuring stiff surfaces such as silicon and metals (https://www.ntmdt-tips.com/, accessed on 29 January 2024). The NSG03 cantilever in Figure 5B had a reported typical spring constant of 1.74 N/m, suitable for analysing slightly more complex or stiffer surfaces, such as many polymer-based surfaces or films. The CSG10 cantilever in Figure 5C has a reported typical spring constant of 0.11 N/m from the manufacturer, tailoring it towards very soft and fragile surfaces such as cells. Each cantilever was used to scan over a single AuNP with similar scan settings. Plots were performed across the centre of each AuNP scanned with a different spring constant cantilever. The height and width were measured on AuNPs in the same sample (Figure 5D). The AuNP measured with an NSG03 cantilever maps the expected spherical shape. The selection of the appropriate tip is a crucial factor in AFM, serving not only to achieve high-resolution imaging but also to minimise anomalies and ensure consistent data sets [65]. The NSG30, NSG03, and CSG10 cantilevers possess a conical tip 500 nm from its apex with a manufacturer-reported half-side angle of 18° ± 2°. The tip’s side angle primarily relates to a measurement anomaly called tip convolution [60], which measures the AuNP at an increased width. Scanning in non-contact topography mode can reduce the tip convolution effect as there is no physical sidewall contact with the conical tip [66]. Cantilevers with a high Q-factor improve image resolution and sensitivity. The Q-factor indicates the amount of energy dissipation and dampening in the cantilever system. A higher spring constant relates to a higher Q-factor, as a high Q-factor oscillates at high amplitude and reduces lateral and torsional bending. The spring constant (ĸ), also known as the force constant, determines the stiffness of the cantilever [67,68].

### 3.9. AuNP Attachment vs. Surface Topography

The topographical resolution of the base substrate is a pivotal factor for the attachment of particles. Generally, as the roughness and spacing between peaks increase, the substrate’s height-to-height correlation function (HHCF) becomes higher. This results in a lower available contact area and fewer contact points, which are influenced by the number, size, and spacing of the contact points between the underlying substrate and the AuNP area. MePPOx z-axis peaks averaged 0.4–0.5 nm, and peak-to-peak distances ranged from 45–80 nm. The AuNP–substrate contact area and number of contacts directly relate to the attachment strength and overall particle stability. The area of contact is primarily a critical factor when the size of the AuNP and its available contact area are close to the magnitude of the substrate’s HHCF. Measured over three 10 × 10 µm regions, the mean HHCF for MePPOx substrates without AuNPs was 0.8 nm². A 2D plot comparing the topography of the MePPOx substrate to the size of the nanoparticle is shown in Figure 6 to display the non-uniformity and adhesion limitations of the contact. 

### 3.10. AuNP Wobbling

AFM topographical scanning occurs in both horizontal (*x*-axis) directions as the scan progresses vertically, which means that surface features are contacted by the cantilever tip from both the left and the right in the fast scan *x*-axis. The stiffer AuNP attached to a low-elastic-modulus surface is expected to display at least a small range of movement due to the force from tip contact, which can be observed in real time by the active trace–retrace scan plot, of which an example is displayed in Figure 7A,B. The effect is also seen in separate trace and retrace images where only one *x*-axis scan direction is shown, Figure 7C. Represented as a ‘blurring’ of the AuNP’s lateral edge on the side of tip contact, a 2D plot maps the effect of this AuNP substrate-attached movement, Figure 7D.

### 3.11. AuNP Movement

Pushing NPs with an AFM cantilever is not new. NPs have been moved across stiff materials and thin film surfaces [69,70,71]. When the cantilever tip laterally contacts an NP in contact mode, there are generally two possible actions. In the first action, the tip contacts the side of the NP, and as it moves up the feature, friction increases until the tip reaches near the top of the NP, where the friction reduces. The second action is at a high enough Set Point force that the NP–substrate bonds are broken, and the NP moves across the surface. This will see a similar rise in friction on NP contact as per the first action. Observation of the scanned image verifies which action has occurred. Contact mode interaction between the cantilever tip and the substrate-bound NP highly depends on the Set Point, the cantilever’s spring constant, and the NP’s adhesion strength to the substrate. Generally, the greater the NP’s contact area, the higher the adhesion to the underlying substrate due to increased bonds. The adhesion area can be challenging to quantify in contrast to the MePPOx surface’s HHCF and the AuNP roughness, which may lower the number of binding sites [72]. This level of roughness and HHCF can limit surface attachment to around 1–4 substrate peaks contacting the AuNP. Covalent bonds are only a few Angstroms in length [73], meaning covalent attachment only occurs at the surface peaks. Weaker forces, such as electrostatic and van der Waals, contribute to AuNP bonding at the substrate as they can influence bonding at greater distances [74]. Therefore, the adhesion of the AuNP to the MePPOx substrate will consist of a minimal area of strong covalent bonding combined with weaker, longer-range forces.

Below the threshold force of breaking the bond that attaches an AuNP to the substrate, the AuNPs can be observed to be pushed along the *x*-axis by the tip while they remain bound to the substrate. This effect is primarily due to the elastic deformation of the MePPOx polymer and is not expected to occur on stiffer substrates [75]. Figure 7A,C show that this movement in line with the scan direction often displays a shadow effect on the initial tip contact side of the AuNP. Although this effect can be seen generally on contact mode scans as a tip convolution effect, a double climbing effect can be observed over some NPs, such as in Figure 7A,B, often those with a smaller NP–substrate contact area. On contact, the tip moves up the AuNP with a significant increase in friction force until a leverage point is reached that pushes the particle laterally, which relaxes the friction force briefly, as shown in the horizontal step in Figure 7D. Once the AuNP moves to a lateral maximum, the friction force increases again as the tip continues moving up and over the structure. Figure 7D shows this ‘double climbing’ effect in a 2D plot of the marked area next to the AuNP in Figure 7C. This effect is not to be confused with what has been termed the Wile E Coyote effect [76], which may occur as the tip moves off sharp, high-surface steps, especially for fast-running scans. However, the double climbing effect occurs when the cantilever tip is moving onto a high-surface step. This effect can be reduced by decreasing the Set Point in contact mode.

Stiff NPs on softer substrates may cause deformation of the underlying surface, increasing the contact area and creating more points of covalent bonding. Plastic deformation of the substrate by the NP can be used to measure the contact area depending on the processes of initial adhesion. The area of permanent plastic deformation is expected to be slightly smaller than the actual deformation caused by the NP, as elastic deformation is not included. NP diameter concerning the adhesion area is related by a simplified formula, Equation (1), with *R* being the NP radius [77].
(1)$$A=\pi R^{2}{SIN}^{2}\theta$$

The contact area between AuNP and MePPOx substrate.

Figure 8A,B show an indent in the MePPOx surface after the tip-induced removal of an AuNP observed between the before and after contact mode scans. Surface area calculations of these indented areas using ImageJ software were roughly 572 nm^2^ and 908 nm^2^ for Figure 8C,D, respectively, which equates to contact diameters of 24 and 30 nm, a large area of contact for a 68 nm diameter spherical particle. The area can be compared to that of a neighbouring AuNP to the left of the indented region in Figure 8C.

AuNP movement occurred during contact mode scanning, where the cantilever tip is in constant contact with the sample surface. Two distinct representations can recognise NP movement. The first is the ‘cut’ particle effect that appears during scanning. This effect relates to NP motion in the fast scan direction. The second effect appears as drag lines in the substrate over gradual movement in both the slow and fast scan direction [69,71]. Depending on the interaction between the cantilever tip and the substrate, a minimum force threshold exists at which NPs will remain immobile. Once this minimum force, controlled by the Set Point, is exceeded, NPs can be observed to move on the surface. In the scenario where all NPs possess uniform size, surface bonding, and geometry, it is anticipated that all AuNPs would exhibit movement above a certain force threshold. Experimentally, this was not the case. Due to surface defects, variations in the geometry, particle placement, roughness gradients, and different particle–surface binding values, not all AuNPs will begin moving at the one applied Set Point force. 

### 3.12. AuNP–Substrate Indentation

Applying a force on the AuNPs greater than the tip–substrate attachment force resulted in AuNP detachment. An NP on a stiff surface would be expected to leave negligible residue or damage at the initial attachment point [78,79]. However, a stiff NP on a deformable surface, such as the MePPOx film, creates a plastic indentation of the initial area of contact, which can be observed after the tip moves the AuNP.

### 3.13. AuNP Movement During Scanning

Breaking the AuNP-MePPOx bonds allows the AuNP to be moved freely across the surface by the cantilever tip during contact mode scanning. An effect that can be recognised during scanning is the sudden disappearance of the AuNPs as the slow scan direction progresses. As shown in Figure 9A, two AuNPs are displaying a ‘cut’ appearance. The ‘cut’ region shows the point where the force applied by the cantilever tip overcomes the AuNP’s adhesion to the surface and moves the AuNP to a new area. Scanning the following line in the *x*-axis, the AuNP appears to have disappeared. Frequently, the AuNP is displaced to the outermost boundary of the scanning area, rendering it unobservable until a broader region is scanned using AM mode. AuNPs in the same image do not display this ‘cut’ feature, meaning insufficient force has been applied to break their bonding to the surface. This effect has been previously reported and can be observed in Figure 9B [69]. Once the AuNP contact is broken, the AuNP has a lower adhesion force to the substrate than the force provided by the tip. Due to this significant imbalance of forces, the AuNPs are pushed to the lateral edges of the scan area. Figure 9C supports this particle cut-off behaviour with an AM mode scan of the region of AuNPs and the following contact mode lateral deflection scan, which demonstrated the cut-off effect on the movement of all these AuNPs of roughly the same size. Figure 9D,E shows the trace and retrace lateral deflections of the contact mode scan with marked regions plotted in Figure 9F. The arrow signifies the increase in force experienced by the cantilever tip at the point of AuNP movement. The retrace value shows no friction spike as the AuNP was moved in the trace scan direction.

MePPOx, a plasma-derived polymer, lacks a crystal orientation or clear surface patterned orientation that may control the direction of moving AuNPs [8,15]. Due to this amorphous property and the roughness of the substrate, particle shape, and other anomalies, a particle may be moved or even rotated in many directions at small steps on each scan line. Once the initial adhesion is broken, the particle is more straightforward to manipulate by the tip and may be pushed over discontinuous steps, as shown in Figure 10A,B. More chaotic movement of AuNPs across the surface can also occur, leaving plastic deformation observed as ‘drag lines’ over the scanned surface, Figure 10. Figure 10A shows the contact mode lateral deflection trace and retrace scans. The white line in Figure 10B maps over one of these draglines. The AuNP being pushed over the MePPOx substrate by the cantilever tip damaged this film. Plotted in Figure 10C is the result of the AuNP pushing into the substrate, leaving a permanent channel. The MePPOx film on either side of the channel has been retracted, creating raised features. This change in substrate morphology may alter the intended properties of the featured surface. Figure 10D is a schematic of the raster scan movement of the cantilever indicating how an AuNP may be moved in both the *x*-axis (fast scan direction) and the *y*-axis (slow scan direction) by pushing the bottom of the AuNP and moving it horizontally and vertically in the scan area on each raster movement. The movement of a stiff AuNP can significantly damage a less dense surface. Permanent surface indentations can occur between the original attachment and its new position, altering the original substrate topography. 

### 3.14. Point of Movement

The point of contact between the tip and a AuNP sees two opposing static forces, one by the tip, the other by the AuNP bonded to the base substrate. If the force applied by the cantilever tip is higher than the AuNP–substrate bond’s resistance to the tip’s lateral applied force, the bonds are susceptible to breaking, allowing the tip to move the AuNP across the substrate. Figure 11A,B show two AuNPs that have had the bonds broken from the substrate. Movement occurred at the particle cut-off point. Measurements and calculations of friction force were performed at five steps of the AuNP movement event. Before the tip–NP contact, the tip is in contact with the MePPOx substrate. Upon first contact, the tip undergoes torsional deformation as it glides along the side of the AuNP, consequently amplifying the frictional force. Nearing the top of the AuNP, the friction force drops as the tip moves horizontally and vertically. The force increases at the shearing of the AuNP-MePPOx bonds, and static friction becomes kinetic, which is the friction force of the tip pushing the AuNP. Post-movement is when the tip connects with the substrate, prior to tip contact. The initial AuNP movement at tip contact can be measured by the separate trace and retrace lateral deflection scans. Figure 11C is a 3D rendering of these scans created with Gwyddion software of the lateral trace (upper image) and retrace (lower image) of AuNP 1. The raised peak on the left side of the lateral trace rendering is the tip contact to the movement of the AuNP. The drop in friction can be seen as the valley in the rendering. Figure 11D provides a table of the values of this event. 

### 3.15. Static and Kinetic Friction 

Calculating the work of adhesion (*W*) between a AuNP and the MePPOx substrate can be performed using the Derjaguin, Muller, Toporov, Maugis (DMT-M) theory [80]. This theory is used to calculate the contact area of a AuNP (*D**M**T*−*M*) using Equation (2), with $K=\frac{4}{3}{\left[\frac{\left(1-v_{1}^{2}\right)}{E_{1}}+\frac{\left(1-v_{2}^{2}\right)}{E_{2}}\right]}^{-1}$, where *E* and *v* are the elastic modulus and Poisson ratio of the sample and tip, respectively. The value *R* is the radius of the spherical AuNP, and *W* is the work of adhesion, classed as the free energy difference between two states, the MePPOx substrate and the AuNP.
(2)$$A_{DMT-M}=\pi {\left(\frac{2\pi W}{K}\right)}^{2/3}R^{4/3}$$

DMT-M theory of AuNP contact area. 

Equation (2) is rearranged to calculate *W*, where the AuNP radius, the contact area, and the elastic modulus for both the AuNP and the MePPOx substrate need to be known, as well as their Poisson ratios. The AuNP radius can either be equated to the expected theoretical diameter or measured by the scan area in an image to account for tip convolution. In this calculation, the theoretical 68 nm diameter was used. The contact area was averaged from the plastic deformation values acquired in Figure 6. The Poisson ratio for colloidal gold is expected to be the same for nanoporous gold at 0.2 [81] and 0.35 for an oxazoline material [81]. The elastic modulus for MePPOx is referenced at approximately 2.5 GPa [82], and 100 GPa for a colloidal AuNP [83]. The results for *W*, with adhesion areas from Figure 6C,D, are 1.25 J/m^2^ and 2.5 J/m^2^, respectively. Relating *W* to the pull-off force as this force increases (this force increases as a negative value, as opposed to the force applied by the Set Point), the contact area reduces until the tip breaks free of the short-range surface forces. DMT theory states the pull-off force in Equation (3) as a spherical cantilever tip on a flat substrate. The initial static force to initially move the AuNP is calculated as $-2.67\times 10^{-7}$ and $-5.34\times 10^{-7}$ Nm using the 1.25 J/m^2^ and 2.5 J/m^2^ values, respectively. Set Point, scan velocity, and torsional spring constant of the cantilever are the primary factors of this initial static force.
(3)$$F_{s}=-2\pi RW_{12}$$

DMT theory of the pull-off force. 

Characterisation of the force required for the cantilever tip to move a AuNP depends on the particle’s size, shape, adherence to the base substrate, roughness, hydrophobicity, and other friction properties between the substrate and particle. Figure 9 focuses on two separate AuNPs on the MePPOx surface that have been moved in the *x*-axis during the scanning process. Figure 9B focuses on one of these AuNPs with a 3D-rendered image via Gwyddion software. A prominent friction peak is apparent at the initial tip–AuNP contact region on the substrate as the scan moves from left to right. A significant increase in friction builds as the tip climbs up the side of the particle. A lower friction region appears after the tip moves over the top of the AuNP, measuring lower friction on the AuNP compared to the MePPOx substrate. Friction drops even further as the tip moves ‘downhill’ off the particle. A raised feature present as an individual line scan indicates the *x*-axis AuNP movement increasing friction. Apparent in the trace scan and disappearing in the following retrace scan, this raised line of friction relates to the data measured in Figure 8B. Measurement of this point over the white line (1) in Figure 9A using Gwyddion and Excel software calculates a kinetic friction force movement of the AuNP at 2.82 nN. Subtracting this value from the 2.53 nN friction over the MePPOx substrate provides a kinetic friction value of 290 pN (0.29 nN), the kinetic force required to continue moving this AuNP. For comparison, the kinetic friction measurement of the second AuNP (2) was measured from Figure 9A at 170 pN (0.17 nN).

## 4. Conclusions

The AFM is a powerful and highly adaptable instrument that can be an all-in-one tool for detailed analysis of complex NP surfaces. It is crucial to acknowledge the potential anomalies that may emerge because of the flexibility and variability of the instrument. Certain irregularities can be mitigated by selecting the appropriate cantilever and configuring optimal scan settings that align with each sample’s topographical and nanomechanical attributes. This is particularly crucial for surfaces characterised by increased complexity in topography, chemical composition, or both. AuNPs creating a functionalised surface have been explored for many applications. Displacement of these AuNPs may affect surface-based functionality to the point where it does not perform as intended. For any novel functionalised surface, physical nanomechanical testing should be performed to verify the limitations of the applied structures with their intended use. 

Initial nanomechanical analysis of the MePPOx describes low roughness, high adhesion, and reasonably high friction. Attached AuNPs to this lower-modulus substrate saw notable anomalies often absent from more straightforward, more controlled surfaces. The anomalies were mainly due to the movement of the AuNPs. Polymers and functionalised surfaces are becoming increasingly interesting in biological and medical devices, meaning that nanomechanical testing of physical properties and limitations must be addressed. AFM testing of this functionalised surface using AFM topography, force curve, and LFM modes provided data on the notable increase in adhesion and lowering of the elastic modulus of the MePPOx substrate at a 20 nm thickness. More critical was manipulating the AuNPs at a reasonably low applied lateral force from the AFM cantilever. Even at meagre forces, AuNPs could be seen to shift relatively quickly on the MePPOx. Applying a slightly higher force can break an individual AuNP’s bonds, allowing the cantilever tip to push the AuNP along the x-axis or in gradual steps over the length of the scan. The minimum threshold of force to move an AuNP was dictated by many factors, each of which varied considerably. The work of adhesion and the force required to move a single AuNP kinetically was calculated for two AuNPs at forces of −2.67 × 10^−7^ Nm and −5.34 × 10^−7^ Nm, and work of adhesion values of 1.25 J/m^2^ and 2.5 J/m^2^, respectively. These weak forces indicate a lack of stability between the AuNPs and the surface, potentially limiting their applicability in specific applications.

## Figures and Tables

**Figure 1 nanomaterials-14-01275-f001:**
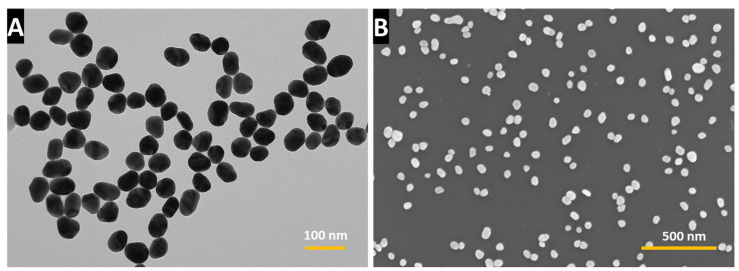
(**A**) TEM micrograph of AuNPs, and (**B**) SEM micrograph of AuNP-coated surface.

**Figure 2 nanomaterials-14-01275-f002:**
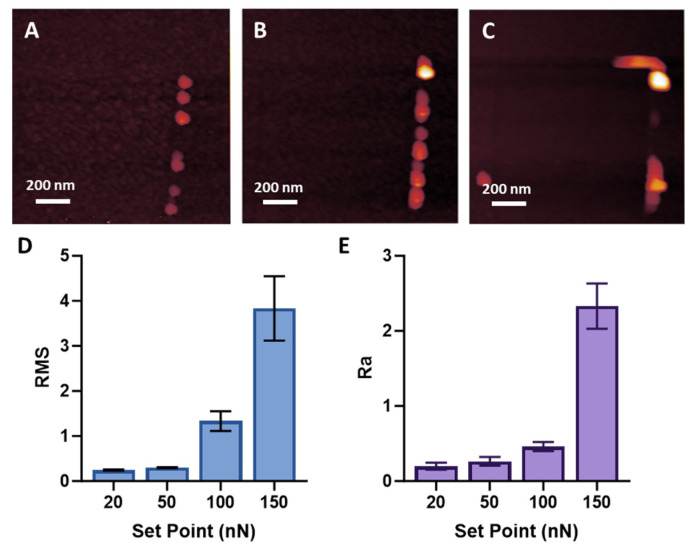
AM mode scans post-contact mode imaging at Set Point forces of (**A**) 100 nN, (**B**) 250 nN, and (**C**) 500 nN, scale bars represent 200 nm. (**D**) RMS and (**E**) Ra of post-AM mode scans concerning applied Set Point. The findings are presented as the mean ± SD, based on a sample size of *n* = 5.

**Figure 3 nanomaterials-14-01275-f003:**
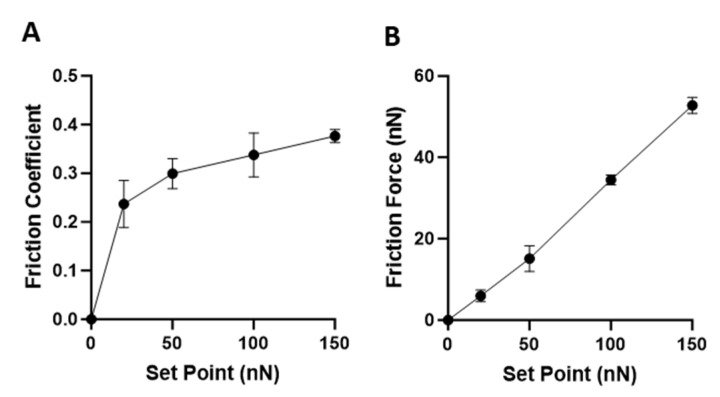
(**A**) The Friction coefficient is derived from contact mode scanning of the Lateral Deflection (the torsional twisting of the cantilever) in both scanning directions and measured by the median value. The friction coefficient and (**B**) friction force versus Set Point are displayed as mean ± SD, with a sample size of *n* = 10.

**Figure 4 nanomaterials-14-01275-f004:**
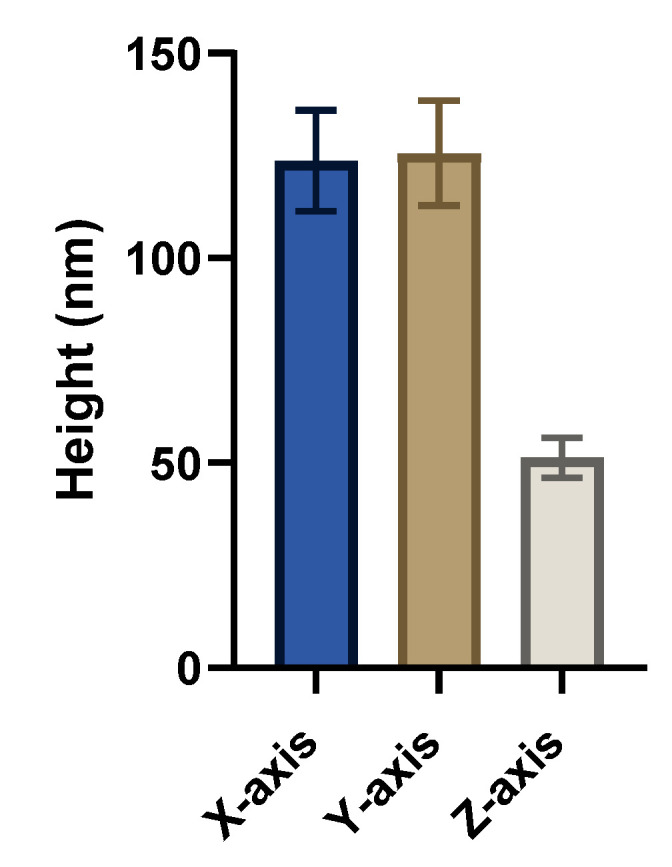
Dimensions of the AuNP nanoparticles were measured in tapping mode on the MePPOx surface in the x, y, and z-axis. Results are displayed as mean ± SD with a sample size of *n* = 10.

**Figure 5 nanomaterials-14-01275-f005:**
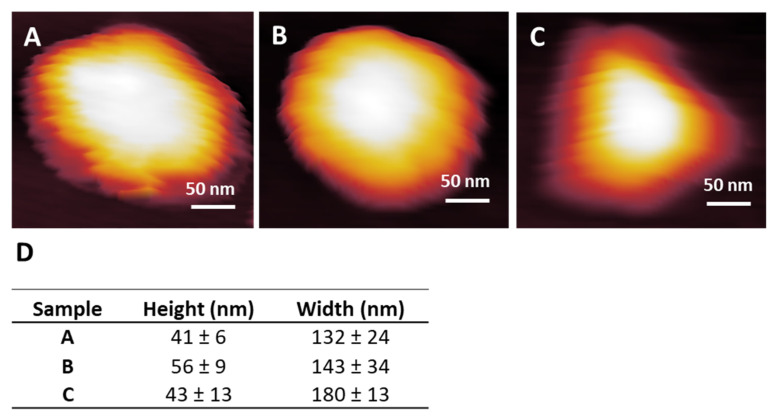
An individual 68 nm AuNP, (**A**) 3D Gwyddion software rendered image using an NSG30 cantilever with a spring constant of 33.4 N/m, (**B**) using an NSG03 cantilever with a spring constant of 3.5 N/m, and (**C**) using a CSG10 cantilever with a spring constant of 0.14 N/m. (**D**) Vertical measurements across the nanoparticles (**A**–**C**), data presented as mean ± SD, *n* = 3.

**Figure 6 nanomaterials-14-01275-f006:**
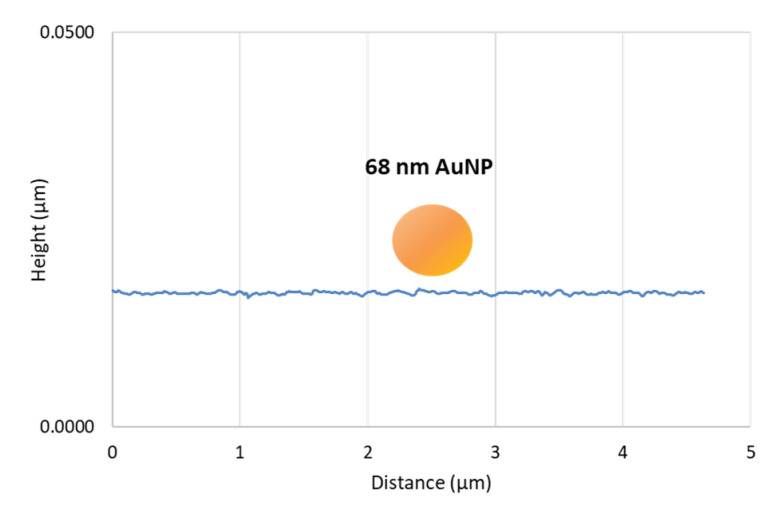
This 2D plot, meticulously constructed using Gwyddion software, is a crucial visual aid in our study. It provides a scaled representation of the MePPOx substrate’s roughness and the complex field HHCF. A visual representation of the 68 nm AuNP is included, which is crucial for understanding AuNP’s behaviour on the MePPOx substrate. The Set Point value was 40 nN.

**Figure 7 nanomaterials-14-01275-f007:**
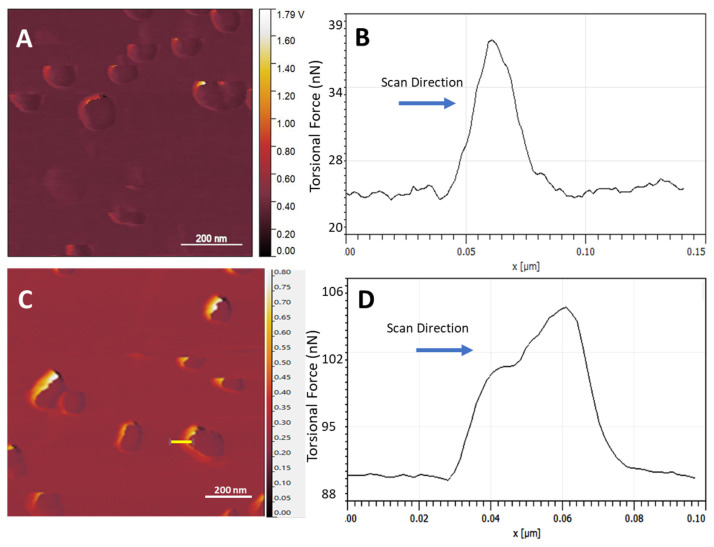
(**A**) Lateral deflection trace scan at a 10 nN Set Point (the yellow line indicates the site where plot (**B**) was executed), (**B**) 2D plot over an individual AuNP, (**C**) lateral deflection trace scan at a 20 nN Set Point (the yellow line indicates the site where plot (**D**) was executed), (**D**) 2D plot over an individual AuNP displaying the double climbing effect.

**Figure 8 nanomaterials-14-01275-f008:**
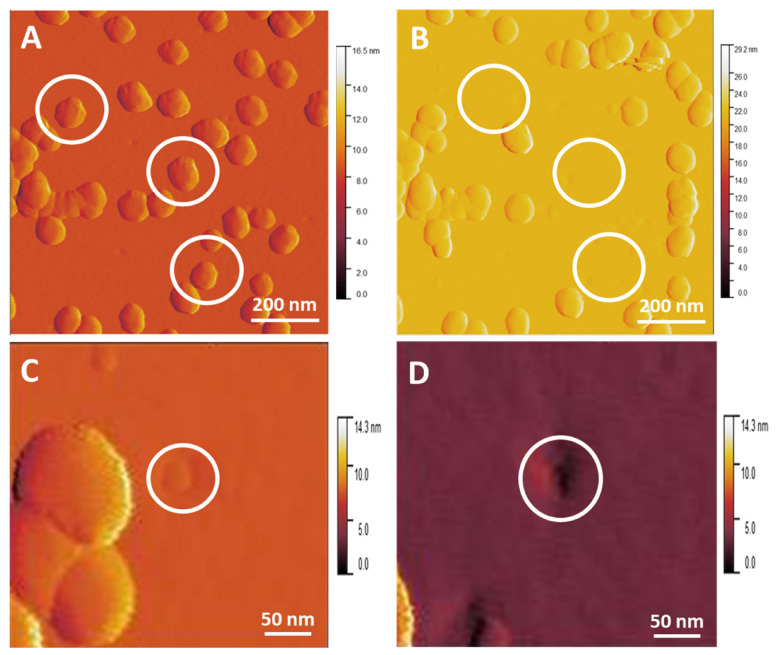
Amplitude image from AC mode scanning with a cantilever spring constant of 2.6 N/m, a free amplitude of 7.9 × 10^−15^ m, and a Set Point of 23.2 nN. The free amplitude output from the instrument’s software was known to be below the system’s noise level, measuring in the femtoscale. This value was confirmed with testing of other cantilevers. (**A**) Before contact mode movement of AuNPs, with three AuNPs highlighted by white circles, (**B**) post-AuNP movement, with the white circles in the same position as the surface in (**A**). The AuNPs have been shifted by the cantilever tip scanned in contact mode (**C**,**D**). These are magnified regions highlighted by white circles showing permanent indentation in the MePPOx substrate after removing the AuNP.

**Figure 9 nanomaterials-14-01275-f009:**
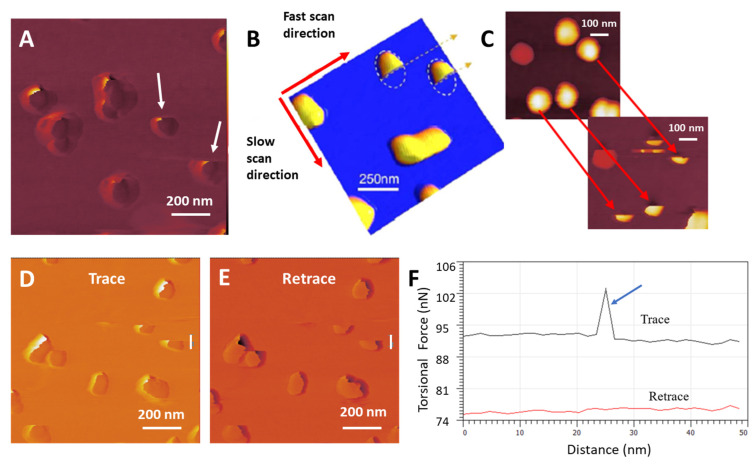
AuNP movement is displayed by ‘cut’ particle appearance, Set Point 40nN. (**A**) Lateral deflection trace scan of AuNPs on MePPOx, and the two white arrows show the cut-off particles. The white arrows highlight the two AuNPs that exhibit the cut-off effect. (**B**) Sb particles on HOPG [69]. (**C**) AuNPs in contact mode before and during their interaction with the cantilever tip. The red arrows follow before and during the cut-off of three of the individual AuNPs, comparing the AM mode scan of the AuNP surface to the subsequent contact mode scan confirming the ‘cut’ appearance and AuNP movement. The yellow dotted lines around the two cut-off AuNPs, show the actual size of the NPs if they had not moved. In lateral deflection (**D**) trace and (**E**) retrace scans, the white vertical lines highlight where the 2D plot was conducted. (**F**) A 2D plot over the region of AuNP movement. The blue arrow signifies the increase in force experienced by the cantilever tip at the point of AuNP movement.

**Figure 10 nanomaterials-14-01275-f010:**
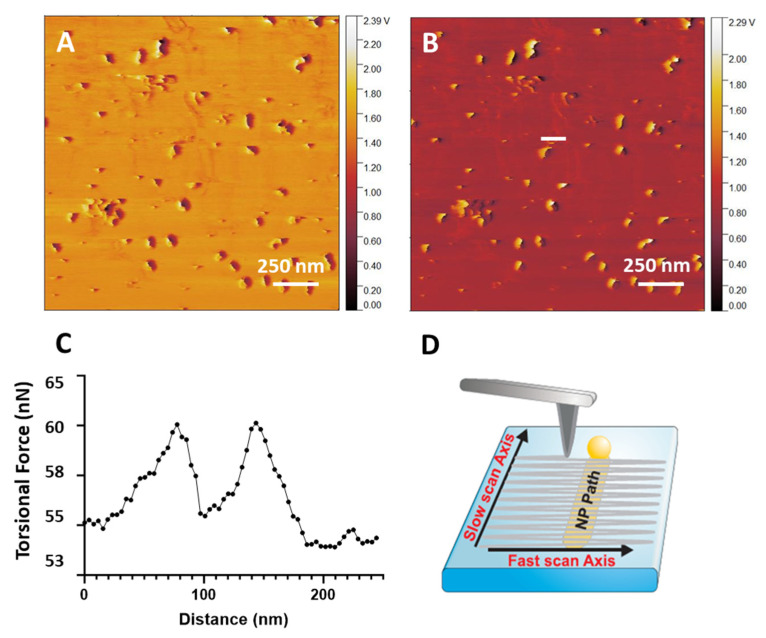
(**A**) LFM lateral deflection trace and (**B**) retrace images showing the two-axis movement of AuNPs on MePPOx over a 2 × 2 µm scan with a Set Point of 40 nN. The white line highlights the location where the scan was conducted. (**C**) A 2D plot across the ‘drag lines’ in (**A**), and (**D**) illustration of the tip’s raster scan motion moving a AuNP in the slow axis.

**Figure 11 nanomaterials-14-01275-f011:**
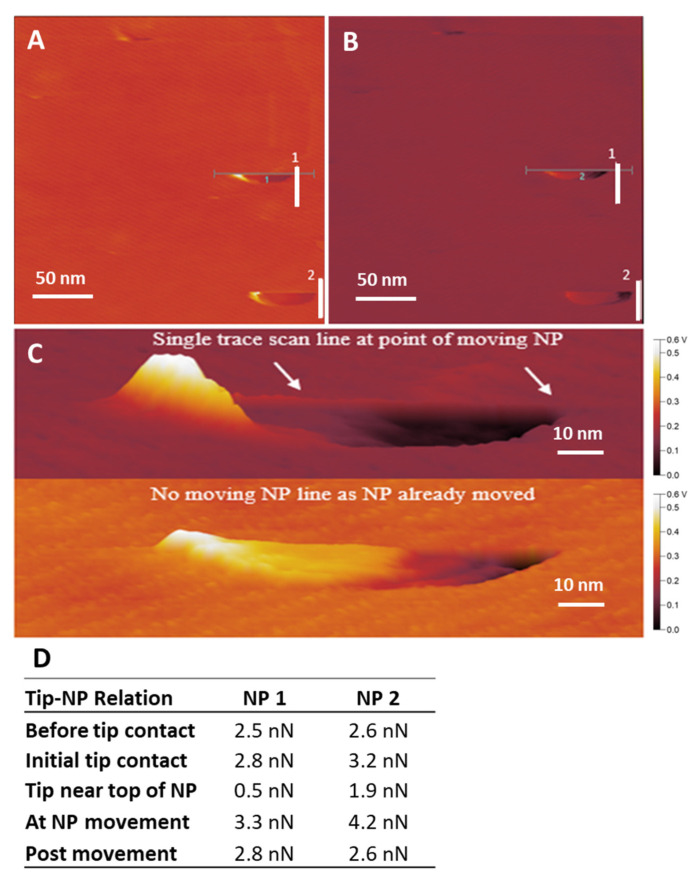
(**A**) Trace and (**B**) retrace lateral deflection scan images displaying the point of AuNP movement. The vertical white line at the end of the cut-off AuNPs is where the post-movement part of the event was measured, and the blue numbers represent the order of the measurement for the two AuNPs measured (**C**) Trace and retrace lateral deflection 3D images of the AuNP movement region. The two white arrows mark either side of the AuNP’s size. (**D**) Friction force values of the movement event of both AuNPs.

**Table 1 nanomaterials-14-01275-t001:** AM and contact topographical mode roughness values of the base MePPOx substrate using an NSG03 model cantilever with a 10 nm radius conical tip and a spring constant of 2.56 N/m. Data presented as mean ± SD, *n* = 3.

Scan Mode	Scan Parameters	Ra (nm)	RMS (nm)
AM Mode	Drive Amplitude 3.1 nm	0.33 ± 0.10	0.70 ± 0.16
AM Mode	Drive Amplitude 10.8 nm	0.29 ± 0.08	0.69 ± 0.19
Contact Mode	Set Point 20 nN	0.16 ± 0.07	0.33 ± 0.01
Contact Mode	Set Point 50 nN	0.14 ± 0.05	0.18 ± 0.0

**Table 2 nanomaterials-14-01275-t002:** Force spectroscopy at 1 and 100 nN of elastic modulus and adhesion values of a 10, 15, and 20 nm thick MePPOx film on a glass coverslip. Results are displayed as mean ± SD, with a sample size of *n* = 3.

Set Point Force		10 nm Film	15 nm Film	20 nm Film
1 nN	Elastic modulus (MPa)	91.7 ± 5.6	88.7 ± 16.0	23 ± 8.3
	Adhesion (nN)	19.3 ± 9.8	21.3 ± 12.5	118.8 ± 28.5
100 nN	Elastic modulus (MPa)	84.8 ± 19.5	79.8 ± 12.4	36 ± 6.6
	Adhesion (nN)	19 ± 3.9	22.4 ± 7.4	71.5 ± 26.6

## Data Availability

The raw data supporting the conclusions of this article will be made available by the authors upon request.

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
