# Peer review of "Exploring the Challenges of Characterising Surface Topography of Polymer–Nanoparticle Composites"

_nanomaterials, 2024, doi:10.3390/nano14151275_

Round 1

Reviewer 1 Report

Comments and Suggestions for Authors

This article focuses on the nanomechanical testing of surfaces containing nanoparticles. It proposes the use of Atomic Force Microscopy (AFM) for nanomechanical testing of material surfaces and provides a detailed analysis of the structured surface composed of poly-methyl oxazoline (MePPOx) films functionalized with colloidal gold nanoparticles (AuNPs). The study specifically investigates the topographical, frictional, adhesion, and wear parameters of the substrate structure and addresses its potential limitations as a biomaterial. However, the manuscript would benefit from the following improvements:

1.       The quality of the figures in the manuscript needs significant improvement, including alignment, positioning, font, font size, and overall image quality enhancement.

2.       Including experiments with different thicknesses of MePPOx films is recommended to better understand the influence of film thickness on nanomechanical properties.

3.       More detailed tests and analyses of the adhesion between AuNPs and the substrate should be considered to deepen the understanding of their stability.

4.       While the potential biomedical applications of MePPOx-AuNP composites are mentioned, it would be beneficial to further elaborate on specific application scenarios and future research directions.

5.       The discussion section would benefit from a more detailed exploration of the limitations of the experiments and potential improvements, such as enhancing AFM resolution or employing other characterization techniques.

6.       The abstract and introduction sections do not clearly highlight the innovative aspects of the work. It is suggested to explicitly state the novel contributions and significance of the study in these sections to improve clarity and impact.

Comments on the Quality of English Language

The manuscript is generally clear, but could benefit from simplifying some complex sentences to enhance readability. Additionally, the text contains minor grammatical errors. Overall, the quality of the English language is good, but some refinements are needed to enhance clarity and correctness.

Author Response

Reviewer 1.

This article focuses on the nanomechanical testing of surfaces containing nanoparticles. It proposes the use of Atomic Force Microscopy (AFM) for nanomechanical testing of material surfaces and provides a detailed analysis of the structured surface composed of poly-methyl oxazoline (MePPOx) films functionalized with colloidal gold nanoparticles (AuNPs). The study specifically investigates the topographical, frictional, adhesion, and wear parameters of the substrate structure and addresses its potential limitations as a biomaterial. However, the manuscript would benefit from the following improvements:

  1. The quality of the figures in the manuscript needs significant improvement, including alignment, positioning, font, font size, and overall image quality enhancement.

We sincerely appreciate your suggestions for improving our manuscript. We have enhanced the image quality, including better alignment, positioning, font selection, font size adjustment, and overall image clarity.

  1. Including experiments with different thicknesses of MePPOx films is recommended to better understand the influence of film thickness on nanomechanical properties.

We measured three film thicknesses in 3.3. Base Film Deformation vs. Thickness, see Table 2 (L238).

  1. More detailed tests and analyses of the adhesion between AuNPs and the substrate

Details are provided in Section 2.1, Sample Preparation, and are discussed throughout the paper. We have attempted to analyse the substrate to the best of our abilities. Furthermore, we could find more detailed references that would further expand the knowledge presented in the paper.

  1. While the potential biomedical applications of MePPOx-AuNP composites are mentioned, it would be beneficial to further elaborate on specific application scenarios and future research directions.

We have added further information to elaborate on specific applications and future research applications (L51)

  1. The discussion section would benefit from a more detailed exploration of the limitations of the experiments and potential improvements, such as enhancing AFM resolution or employing other characterization techniques.

We have expanded the discussion to include limitations and suggestions to improve resolution (L501).

  1. The abstract and introduction sections do not clearly highlight the innovative aspects of the work. It is suggested to explicitly state the novel contributions and significance of the study in these sections to improve clarity and impact.

Thank you for this valid comment. We have added further information to the abstract (L20) and the introduction (L51).

The manuscript is generally clear but could benefit from simplifying some complex sentences to enhance readability. Additionally, the text contains minor grammatical errors. Overall, the quality of the English language is good, but some refinements are needed to enhance clarity and correctness.

Thank you. We have made grammatical improvements throughout the manuscript to enhance readability.

Reviewer 2 Report

Comments and Suggestions for Authors

The manuscript reports an AFM-based study on the structural and mechanical properties of gold nanoparticles deposited on a polymeric film. To that effect, the manuscript reports data on the topography of the gold NPs, the adhesion and friction of the NPs on Poly-methyl oxazoline (MePPOx) substrates, and more extensively on the lateral displacement of the NPs.  Several AFM modes are used in this study, tapping mode AFM for topographical imaging, contact mode AFM for the displacement of the tip, nanoindentation to measure the elastic modulus of the polymer and lateral force microscopy for measuring friction coefficients. Most of the manuscript is devoted to discuss the displacement of the NPs by the AFM tip.

It was hard to read the manuscript. The scientific goals of the manuscript are not clear. There is a wealth of information but the manuscript lacks a coherent structure. The figures are of an uneven quality. Some of them are good but others of a mediocre quality (see below). I recommend a major revision. The authors should specify with clarity the specific goals and the new knowledge provided by the results.  

1 Topography is obtained by using two different modes, tapping mode AFM and contact AFM. For each figure, the manuscript should specific the imaging mode and the parameters used for the mode (free and set point amplitude for tapping mode) and applied force for contact AFM.

2 I would suggest to change the sentence in page 3 lines 119-123: ‘All AFM data

and surface roughness calculations were performed in Amplitude Modulation (AM) tapping and contact topography mode’ by ‘ All AFM data and surface roughness calculations were performed by either  Amplitude Modulation (AM) tapping or contact AFM’.

3 The manuscript should include some general references about tapping mode AFM and force spectroscopy. Suggestions: R. Garcia, R. Perez, Surf. Sci. Rep.  2002, 47, 197-301 and Y.F. Dufrene et al. Nat. Nanotechnol. 2017, 12, 295-307.

4 The title has some unjustified grandeur. I would suggest to remove the words ‘Unravelling the Complexity:’

5 The manuscript should include a figure on the force spectroscopy measurements to determine the elastic modulus of the polymer.

6 Figure 7. Some panels are in units of volts. The authors should convert them to nm. Similarly, for figures 10-11).

7 Figure 8. The colour bars in panels C & D are distorted.

8 Figure 9. Scale numbers in 9A ara hard to read.

Author Response

Reviewer 2

The manuscript reports an AFM-based study on the structural and mechanical properties of gold nanoparticles deposited on a polymeric film. To that effect, the manuscript reports data on the topography of the gold NPs, the adhesion and friction of the NPs on Poly-methyl oxazoline (MePPOx) substrates, and more extensively on the lateral displacement of the NPs.  Several AFM modes are used in this study, tapping mode AFM for topographical imaging, contact mode AFM for the displacement of the tip, nanoindentation to measure the elastic modulus of the polymer and lateral force microscopy for measuring friction coefficients. Most of the manuscript is devoted to discuss the displacement of the NPs by the AFM tip.  

It was hard to read the manuscript. The scientific goals of the manuscript are not clear. There is a wealth of information but the manuscript lacks a coherent structure. The figures are of an uneven quality. Some of them are good but others of a mediocre quality (see below). I recommend a major revision. The authors should specify with clarity the specific goals and the new knowledge provided by the results.   

1 Topography is obtained by using two different modes, tapping mode AFM and contact AFM. For each figure, the manuscript should specific the imaging mode and the parameters used for the mode (free and set point amplitude for tapping mode) and applied force for contact AFM.

The authors sincerely appreciate your insightful comments and suggestions, which have significantly strengthened this manuscript. This is a valid point. We have changed from tapping mode to AM mode and added the spring constant, free amplitude, and set point values.

The mention of cantilever amplitude, free air, etc., is as follows:

-table 1 (L210) Spring constant and drive amp are mentioned

-fig 2 spectroscopy (L334)

-fig 3 contact mode, no amplitude(L353)

-fig 5 spring constant (L431 and 436)

-fig 6 do not need amplitude, Set Point (L456)

-fig 8 done, has an amplitude line( L530)

-fig 9 AM mode, set point of 40 nN (L577)

-fig 10 contact mode no amplitude, set point 40 nN, (L608)

2 I would suggest changing the sentence in page 3 lines 119-123: ‘All AFM data

and surface roughness calculations were performed in Amplitude Modulation (AM) tapping and contact topography mode’ by ‘ All AFM data and surface roughness calculations were performed by either  Amplitude Modulation (AM) tapping or contact AFM’.

Thank you for your valuable input. We have corrected the sentences to read, “All AFM data and surface roughness calculations were performed by either Amplitude Modulation (AM) mode or contact mode” (L132).

3 The manuscript should include some general references about tapping mode AFM and force spectroscopy. Suggestions: R. Garcia, R. Perez, Surf. Sci. Rep.  2002, 47, 197-301 and Y.F. Dufrene et al. Nat. Nanotechnol. 2017, 12, 295-307.

We have added some general references about tapping mode AFM and force spectroscopy (L133).

4 The title has some unjustified grandeur. I would suggest removing the words ‘Unravelling the Complexity:’

Thank you for your suggestion; we have changed the title to read “Exploring the Challenges of Characterising Surface Topography of Polymer-Nanoparticle Composites” (L2)

5 The manuscript should include a figure on the force spectroscopy measurements to determine the elastic modulus of the polymer.

 The polymer's Elastic Modulus was measured by force spectroscopy and is listed in Table 2 (L238).

6 Figure 7. Some panels are in units of volts. The authors should convert them to nm. Similarly, for figures 10-11).

 Unfortunately, we could not convert the lateral deflection data from volts to nanometers because the software did not allow an accurate conversion.

7 Figure 8. The colour bars in panels C & D are distorted.

Thank you for pointing out this oversight in Figure 8. We have corrected the scale bar to remove the distortion (L529)

8 Figure 9. Scale numbers in 9A are hard to read.

We have removed the scale bar on the side of image 9A, as it is unnecessary; all AFM images in Figure 9 already include scale bars inset within the images (L577).

Round 2

Reviewer 1 Report

Comments and Suggestions for Authors

The authors have addressed my concerns. The manuscript can be accepted.

Comments on the Quality of English Language

The manuscript is clearly written.

Author Response

Dear Reviewer , Thank you for your comments and suggestions to improve the quality of our manuscript.

Reviewer 2 Report

Comments and Suggestions for Authors

The authors have addressed the previous comments but one.

 1 In several panels,  the y-axis shows a signal in volts. Those signals are related to the lateral force. Signal in volts are specific of a given instrument (sensitivity of the photodiode), therefore those values would not be useful to guide the experiments of  other scientists.  Those panels should be changed to units of force. Please provide details on the calibration procedure. 

 2 The caption of Figure 8 reads a ‘free amplitude of 7.9x10-15 m’. Amplitude values in the fm range are unreachable. Well below the floor noise level of the authors’ instrument (about 50-100 pm).

Author Response

Dear Reviewer , we apologise for missing these important points in our last review, we have now addressed the comments, see below:

The authors have addressed the previous comments but one.

  1. In several panels, the y-axis shows a signal in volts. Those signals are related to the lateral force. Signal in volts is specific of a given instrument (sensitivity of the photodiode), therefore those values would not be useful to guide the experiments of other scientists.  Those panels should be changed to units of force. Please provide details on the calibration procedure.

Response: We have changed all figures on the y-axis to Torsional Force (nN), please see updated figures (Figure 7(L512), 9(L583) and 10(L613)). Details on the calibration procedure have been add in 2.4 AFM Topography, “Cantilever calibration was achieved through a force curve procedure on a glass microscope slide. AFM software fitted the curve’s deflection slope to determine cantilever sensitivity, Q-factor, frequency, amplitude, and spring constant values. Thermal tuning of the cantilever was performed specifically for tapping mode topography” (L127).

  1. The caption of Figure 8 reads a ‘free amplitude of 7.9x10-15 m’. Amplitude values in the fm range are unreachable. Well below the floor noise level of the authors’ instrument (about 50-100 pm).

Response:  A statement has been added to the figure legend. “The free amplitude output from the instrument’s software was known to be below the system’s noise level, measuring in the femtoscale. This value was confirmed with testing of other cantilevers” (L534).